# A simple methodology to detect and quantify wind power ramps

Bedassa R. Cheneka[1], Simon J. Watson[1], and Sukanta Basu[2]

[1]Faculty of Aerospace Engineering, Wind Energy Section, Delft University of Technology, Delft, The Netherlands
[2]Faculty of Civil Engineering and Geosciences, Delft University of Technology, Delft, The Netherlands

**Correspondence:** Bedassa R. Cheneka (b.r.cheneka@tudelft.nl)

**Abstract.**

Knowledge about the expected duration and intensity of wind power ramps is important when planning the integration of wind power production into an electricity network. The detection and classification of wind power ramps is not straightforward due to the large range of events that are observed and the stochastic nature of the wind. The development of an algorithm that can detect and classify wind power ramps is thus of some benefit to the wind energy community. In this study, we describe a relatively simple methodology using a wavelet transform to discriminate ramp events. We illustrate the utility of the methodology by studying distributions of ramp rates and their duration using two years of data from the Belgian offshore cluster. This brief study showed that there was a strong correlation between ramp rate and ramp duration, that the majority of ramp events were less than 15 hours with a median duration of around eight hours and that ramps with a duration of more than a day were rare. Also, we show how the methodology can be applied to a time series where installed capacity changes over time using Swedish onshore wind farm data. Finally, the performance of the methodology is compared with another ramp detection method and their sensitivity to parameter choice is contrasted.

**Keywords.** detection, discrimination, ramps, surrogates, wavelets, wind power

## 1 Introduction

Rapid changes in wind speed can cause ramps in wind power production of a wind farm. With plans to install a large amount of capacity in the North Sea, understanding swings in offshore wind farm power production will become important for wind farm and network operators to manage the integration of wind power into the electricity system. In their development of a wind power variability index, Kiviluoma et al. (2014, 2016) distinguish between three different time-scales which are of importance to system operators. The first of these is sub-hourly up to two hours where load following and frequency control is required. The second and that where wind power generation is stated to cause most ramping is for timescales between one hour and around 15 hours. For these time-scales, there can be significant impact on thermal plant start-ups and shut-downs, though this depends on the characteristics of the plant installed in the particular power system. The final ramping timescales of relevance are between 10 hours and around a day. This is of relevance where large scale storage is required such as hydro pump-storage. An understanding of the prevalence and magnitude of ramps across a range of timescales is thus important.

Wind power ramps are influenced by the dynamics of atmosphere-ocean systems which could be either mesoscale or synoptic-scale. Therefore, meteorological systems that evolve over time play a significant role in the occurrence of power ramps (Marquis et al., 2011). Low-pressure systems, cold fronts, low-level jets, thunderstorm outflows, and dry lines can cause ramp-up (increasing wind) events (Sevlian and Rajagopal, 2013; DeMarco and Basu, 2018) whereas ramp-down (decreasing wind) events occur due to the reduction or reversal of these physical processes (Ferreira et al., 2011). Short-duration (rapid) power ramps are mainly influenced by mesoscale systems, whereas synoptic systems tend to be responsible for longer duration power ramps (Drew et al., 2018).

There is no accepted definition or classification of wind power ramps except that they are manifested in terms of a significant change in production over a relatively short time. The quantification of the duration and magnitude of wind power ramps has been explored by several scholars. Most of the studies set thresholds with respect to the rated power of the wind farm to detect wind power ramps. One such definition (Bossavy et al., 2010; Zhang et al., 2017) defines a ramp as a minimum change in wind farm output $\Delta P$ as a fraction of the rated wind power $P_R$ of the wind farm over a period of time ($\Delta t$). Different researchers consider different rates of change to define ramps, for example, Cutler et al. (2007) define a power ramp when there is a change in wind power production of 75% of $P_R$ within a $\Delta t$ of 3 hours or 65% of $P_R$ within a $\Delta t$ of 1 hour. In contrast, Bossavy et al. (2010) define a wind power ramp when there is a change in wind power of 50% of $P_R$ over one hour. Other researchers such as Bianco et al. (2016) and Gallego-Castillo et al. (2015) use yet different percentage changes in wind power and time ranges to define wind power ramps. There have been studies to detect power ramps without using any pre-defined change in wind power relative to rated power and time. An optimised swinging door algorithm was used by (Zhang et al., 2017) to extract ramps where the ramp definition parameters related to power change and timescale could be easily adapted. An optimal method based on scoring functions (Sevlian and Rajagopal, 2013) was used to detect ramps of varying lengths at a US wind farm. These authors used a piece-wise linear trending fit to remove short-term stochastic fluctuations.

Even though there has been a significant body of work to detect wind power ramps, it is clear that there is no precise consensus as to the definition of a ramp. Indeed, it may be necessary to extract information about a range of power ramp events depending on the requirements of the wind farm operator or the utility as described above. What is required is a robust method which can extract ramps of arbitrary magnitude and duration and to discriminate above the incoherent stochastic noise level. In this paper, we propose an improved method to discriminate ramp events above incoherent stochastic variations using wavelets. Wavelets have been used in the past to extract ramp events from time series, e.g. (Hannesdóttir and Kelly, 2019; Ji et al., 2015; Gallego et al., 2014; Coughlin et al., 2014). We build on this work by demonstrating how a wavelet transform can be used in conjunction with the generation of wind power surrogates to give a robust method for the detection of wind power ramps of varying magnitude and duration. Rather than relying on fixed power or timescale thresholds, the methodology uses a method of discrimination based on statistical thresholds. We illustrate the methodology and its application using data from Belgian offshore and Swedish onshore wind farms. Firstly, we describe the methodology and illustrate its application using a ten-day period of data. Next, the sensitivity of the discrimination of ramps from natural stochastic variation is investigated using a longer period to generate the surrogate distributions. Then, we show the utility of the approach in terms of characterising the distribution of ramp rates, their duration and diurnal-seasonal variation using two years of the offshore wind power data. Next,

the versatility of the methodology is demonstrated for a non-stationary time-series where installed capacity changes over time. Finally, we compare the methodology with another commonly used approach, namely the min-max method (Bianco et al., 2016).

## 2 The wind farm data

The Belgian transmission system operator, Elia, makes available 15-minute power output data for the aggregated fleet of
Belgian onshore and offshore turbines (Elia, 2020). In this work, we have used offshore data over a period of two years from 2015–2016 when the combined Belgian offshore wind power capacity was 712 MW. For simplicity, the 15-minute values were normalised to the total capacity before analysis to create a time series of values $P(t)$. In addition, we make use of Swedish onshore hourly wind power data for the period 2000–2001 aggregated within the SE1 price region (SCB, 2017). The installed capacity in this region increased from about 500 MW to 1300 MW over this period. Further details of this data-set can be found
in EEM20 (2020).

## 3 Wavelet decomposition

The continuous wavelet transform (CWT) can be used to decompose a series of data using a mother wavelet function ($\psi$) by varying its dilation and translation. A mother wavelet function with scale $a$ and position $b$ can be defined as (Mallat, 2009):

$$\psi^{a,b}(t) = \psi\left(\frac{t-b}{a}\right). \tag{1}$$

The CWT $W(a,b)$ of a signal $X(t)$ is produced by the convolution of the mother wavelet function over a range of scales and positions:

$$W(a,b) = \frac{1}{\sqrt{a}} \int_{-\infty}^{\infty} X(t)\psi\left(\frac{t-b}{a}\right) dt. \tag{2}$$

A wavelet transform is thus able to localise the scales of a series of data in time which makes it a useful function to detect and characterise wind power ramps. We use the Daubechies level 1 (Haar) mother wavelet to decompose the time series of
power values. This wavelet is useful to detect abrupt changes in a level which might be expected to occur during a ramp event.
Using values over a ten-day period, 27 Jan 2015 to 07 Feb 2015, taken from the Belgian offshore wind power data, a CWT was applied and the results are shown in Figure 1 comparing the original time series (a) with the corresponding CWT values (b). It can be seen that a high magnitude of $W$ corresponds to a strong power ramp. Similar finding has been reported elsewhere (Gallego et al., 2014; Hannesdóttir and Kelly, 2019). However, what is not clear is what magnitude of $W$ can be considered
as a ramp above the incoherent stochastic variations in wind power. In the following section, we consider how to discriminate ramps above such stochastic variations.

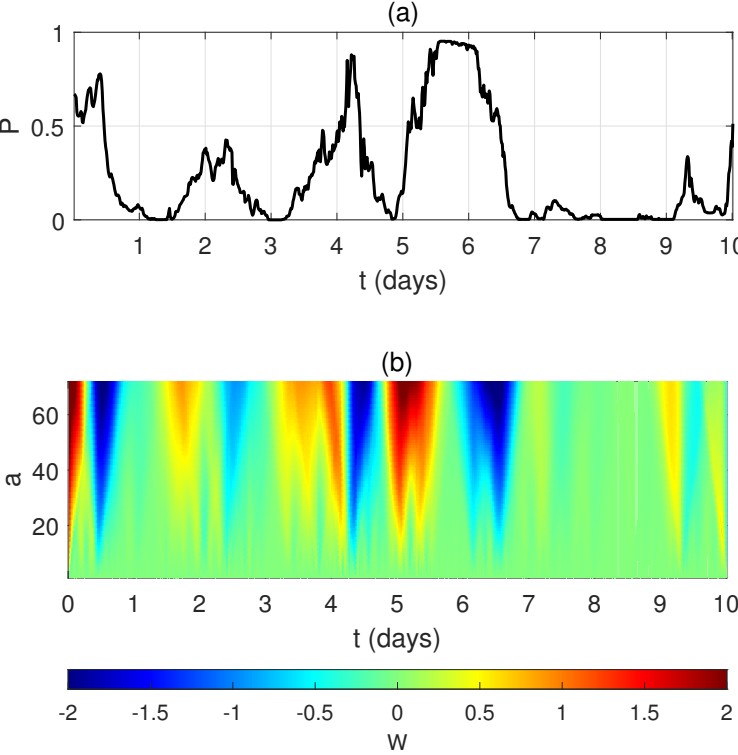

**Figure 1.** (a) aggregated Belgian offshore wind power output over the period 27$^{\text{th}}$ Jan 2015 to 7$^{\text{th}}$ Feb 2015. (b) continuous wavelet transform of the wind power data using the Haar mother wavelet.

## 4 Discrimination of ramp events

Random shuffling is a technique to generate surrogate data from an original time series which preserves limited statistical properties of the original data, namely their distribution. However, it destroys the auto-correlation within a time series. Randomly shuffled surrogates have been used to test for non-linearity in a time series (Theiler et al., 1992). It has also been used to test for stationarity in temporal data (Laurent and Doncarli, 1998; Davy and Godsill, 2002; Borgnat and Flandrin, 2009; Guarin et al., 2010; Borgnat et al., 2010). Furthermore, surrogates have been applied to discriminate gusts and other coherent structures from incoherent noise in high frequency wind speed data (Dunyak et al., 1998; Gilliam et al., 2000).

In Figure 2, as an example, we analyze a surrogate based on the ten-day time series shown in Figure 1(a). Figure 2(a) shows a comparison between the auto-correlation of the original time series of normalised power values, $P(t)$ and that of a randomly shuffled time series of these values, $P^*(t)$. It can be seen that any coherent structure in the original data is destroyed. Figure 2(b) shows the continuous wavelet transform of the surrogate time series, $W^*(t)$. It can be seen that the lower frequency (higher scale value) structure that was seen in Figure 1(b) has disappeared and the power in the transformed wavelet spectrum is much more distributed over all scales.

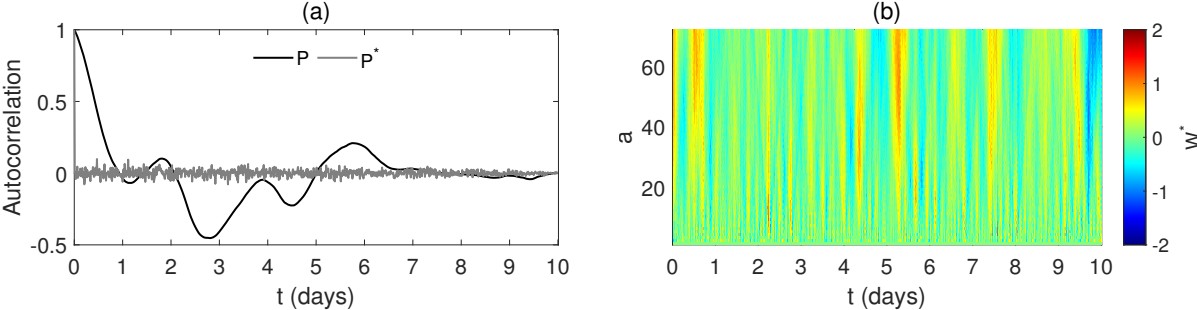

**Figure 2.** (a) the auto-correlations of the original wind power ($P$) and the normalised wind power surrogate ($P^*$); (b) the CWT of the normalised wind power surrogate.

In order to test the hypothesis that the value of a wavelet coefficient represents a ramp event, we generate 100 such randomly shuffled surrogates of normalised wind power, $P_i^*(t)$, where $i = 1$ to $100$, based on the ten days of data. For each surrogate time series, the CWT is generated to give a series of coefficients $W_i^*(a, b)$. These are used to generate distributions of coefficient values (containing $100 \times b$ values) for each scale $a$, against which the CWT coefficient of the original, $W(a, b)$ can be compared. In Figure 3(a)–(d) we show the distributions for $W$ and $W^*$ at the scale $a = 40$ where we discriminate the $W$ values based on the largest 10%, 5%, 2% and 1% of $|W^*|$ values, respectively. The threshold values, $\pm W_T^*$ are shown for each plot in Figure 3.

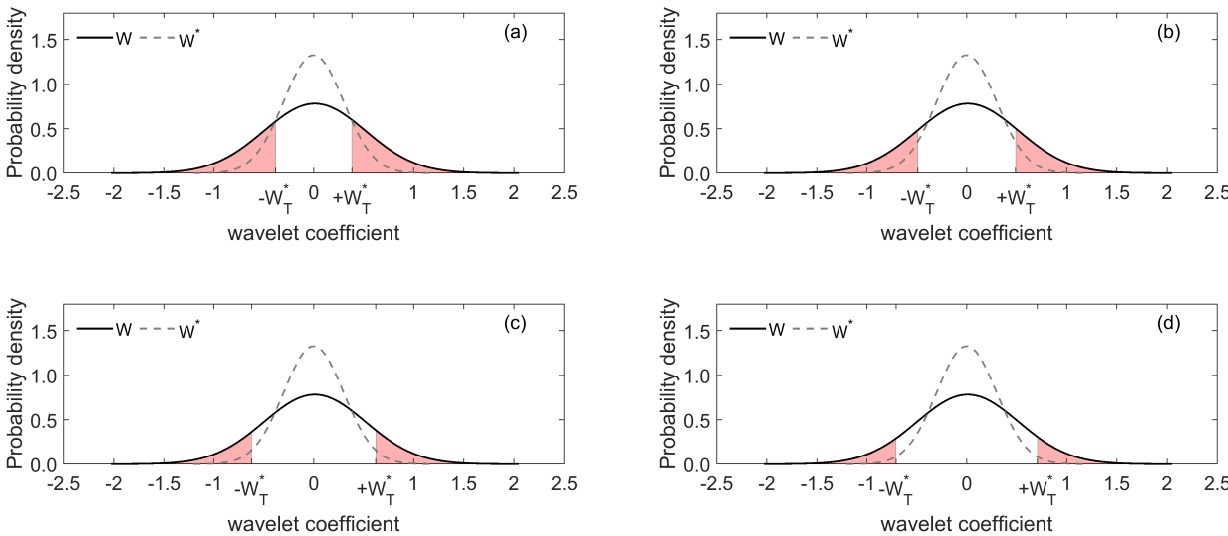

**Figure 3.** The distributions of $W$ and $W^*$ shown at the scale ($a = 40$). The shaded regions show the $W$ values which are classified as ramp events based on thresholds $\pm W_T^*$ set using the the highest (a) 10% (b) 5% (c) 2% and (d) 1% of the $W^*$ distribution.

We then extend this method of discrimination to all the scales of $W$. So, for each scale $a$, we compute a scale-dependent threshold $W_T^*(a)$ for a specific discrimination level by utilising the $|W_i^*(a, b)|$ values from all the surrogates. If the value of

$|W(a, b)|$ is greater than this threshold $W_T^*(a)$, then the null hypothesis (no ramp) is rejected at the specific discrimination level and the event is assumed to be a wind power ramp. We repeat this for the four different discrimination levels used above, namely the 10%, 5%, 2% and 1% levels.

Figure 4 shows the result of using this approach to discriminate the wind power ramps at each scale. The plot is similar to the bottom plot in Figure 1, but now values which do not satisfy the criterion to be considered as ramps have been removed and are shown as white with different null hypothesis testing. Only the colour shaded values that satisfy the requirement to be considered as wind power ramps for different discrimination levels are shown.

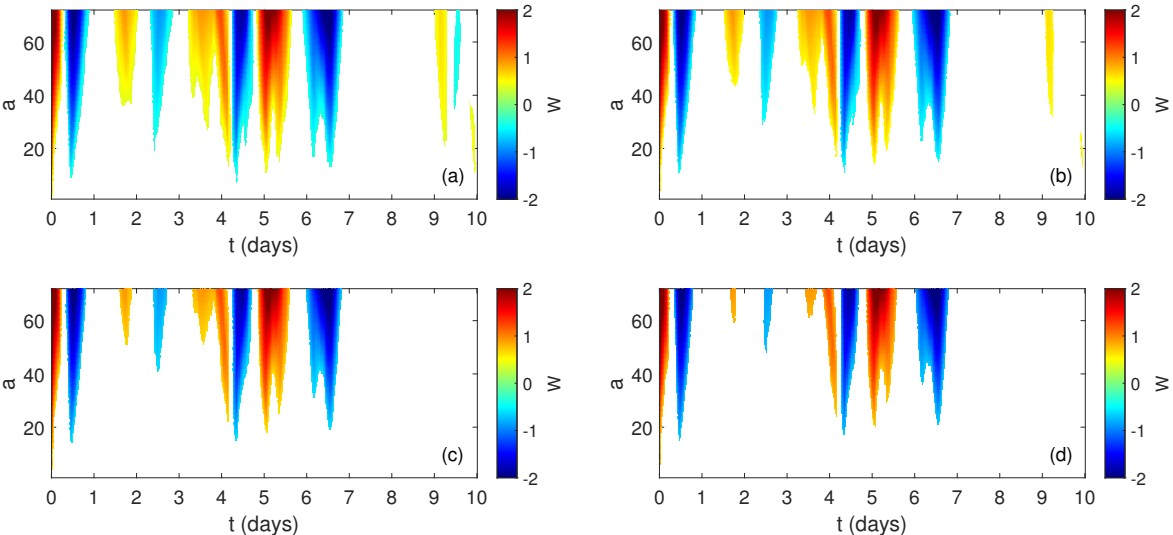

**Figure 4.** The CWT coefficients $W(a, b)$ of the normalized wind power discriminated against the distribution of $W_i^*(a, b)$ using the (a) 10%, (b) 5%, (c) 2% and (d) 1% discrimination levels. The colour scale is blue (ramp-down events), red (ramp-up events) and white (no ramp).

It is then possible to sum $W(a, b)$ over all discriminated scales up to the maximum resolved, $a_{max}$ at each time step, $t = b$ to calculate mean normalised power ramps, $R(t)$:

$$R(t = b) = \frac{1}{a_{max}} \sum_{a=1}^{a_{max}} W_R(a, b) \tag{3}$$

where:

$$W_R(a, b) = W(a, b) \qquad \text{when} \qquad |W(a, b)| \geq W_T^*(a)$$
$$W_R(a, b) = 0 \qquad \text{when} \qquad |W(a, b)| < W_T^*(a)$$

Figure 5 shows the original ten-day time series of wind power values with the normalised ramp values, $R(t)$ superimposed. Power ramps are now clearly defined in terms of both timing and magnitude. Although there is not a large difference in

those events which are classified as ramps, in particular, the events on days nine and ten are excluded at the three highest discrimination levels. For the remainder of the paper we have chosen the 10% level to provide a good balance between the removal of stochastic variation whilst preserving ramp events that would be a relevance from a power system perspective.

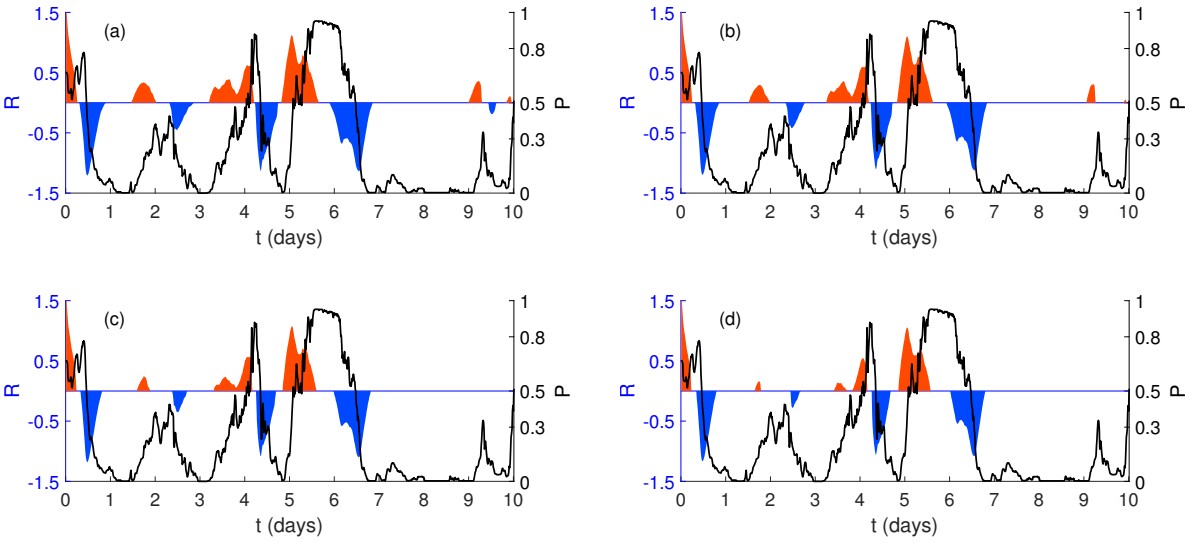

**Figure 5.** Normalised wind power ramps, $R(t)$ is superimposed over the normalised wind power values $P(t)$ for the ten-day period discriminated at the 10% (a), 5% (b), 2% (c) and 1% (d) levels. Ramp-up events are shown in green and ramp-down events in orange.

## 5 Sensitivity to length of surrogate series

To test the generality of the technique, we consider further testing periods and increase the length of time for which the surrogate distributions are calculated. Three additional ten-day periods are selected and for each period, we examine the sensitivity of the results to the length of surrogate, namely: the same ten-day period and one calendar year of values encompassing the ten-day period. These cases are summarised in Table 1.

**Table 1.** Test periods and length of surrogate series used to detect and quantify ramps.

| Test Period | Start | End | Surrogate Period 1 ($S_1$) | Surrogate Period 2 ($S_2$) |
|---|---|---|---|---|
| T1 | 28 Jan 2015 | 7 Feb 2015 | Same as test period | 2015 |
| T2 | 20 Nov 2015 | 30 Nov 2015 | Same as test period | 2015 |
| T3 | 27 Jan 2016 | 5 Feb 2016 | Same as test period | 2016 |
| T4 | 3 Nov 2016 | 13 Nov 2016 | Same as test period | 2016 |

As before, for each case, we generate 100 surrogates and the wavelet coefficients $W(a, b)$ are discriminated against the distributions generated using the two different surrogate periods in Table 1. The results are presented in Figure 6. It can be seen

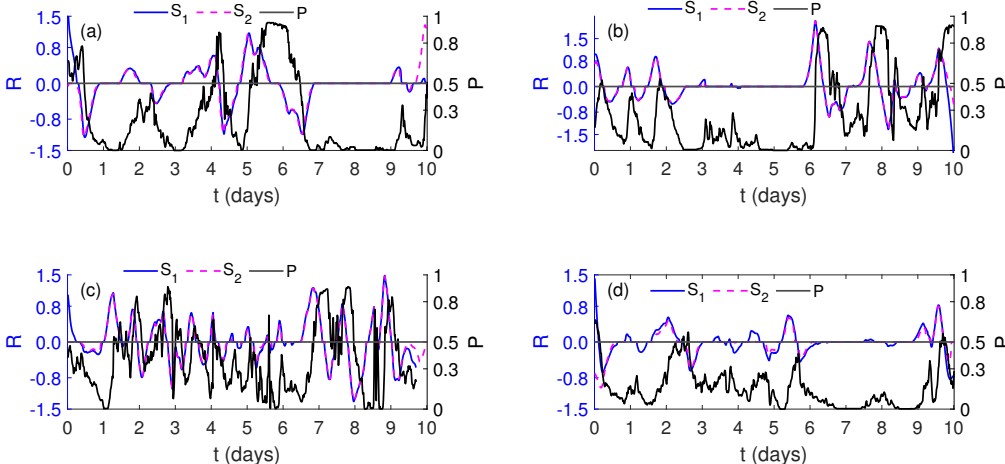

**Figure 6.** Normalised power ramps: (a) period T1; (b) period T2; (c) period T3; and (d) period T4. Normalised wind power is shown in black. The ramp-ups and ramp-downs derived using surrogate statistics from the ten-day period (S1) are shown in blue; and from the one-year period (S2) are shown in purple.

that once again, ramp periods are well discriminated from periods of incoherent stochastic variation. In addition, the results show no difference when using a longer period to generate the surrogates except at the very beginning and end of the time
series. This is due to boundary effects inherent in using a convolution function which is integrated over all time and should thus be disregarded in any comparison. The fact that the results show no differences when using an extended surrogate period confirms that the process is filtering out short-term incoherent fluctuations and that a ten-day period is sufficient to capture these.

## 6 Ramp rates and duration

In this section, we show how the methodology can quantify both ramp rates and their duration, using two years of the Belgian offshore wind power data for 2015–2016. We also use this two-year period to produce the surrogate distributions for deriving the 10% discrimination level. Firstly, we generate a time series of normalised ramp rates. As can be seen in Figures 5 and 6, there are discrete periods of ramp-up and ramp-down events. For each ramp-up period $k$ and ramp-down period $l$ we calculate the average ramp-up rate, $R'_u(k)$ and average ramp-down rate $R'_d(l)$, respectively:


$$R'_u(k) = \frac{\sum_{t=1}^{n_k} R(t)}{D(k)} \tag{4}$$

$$R'_d(l) = \frac{\sum_{t=1}^{n_l} R(t)}{D(l)} \tag{5}$$

where the $n_k$ normalised power ramp-up values $R(t)$ are summed over the duration $D(k)$ of the $k^{\text{th}}$ ramp-up event and the $n_l$ normalised power ramp-down values $R(t)$ are summed over the duration $D(l)$ of the $l^{\text{th}}$ ramp-down event.

## 6.1 Overall distributions

Distribution plots of ramp rates over the entire two year period as a function of duration (binned by hour) are shown in Figure 7. The ramp-up and ramp-down event distributions are broadly similar in nature though there are some features of note:

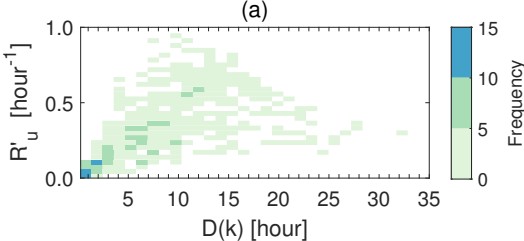 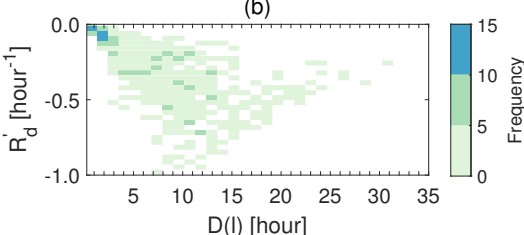

**Figure 7.** Distributions of normalised ramp rates as a function of duration for: (a) up-ramp rates, $R'_u(k)$ and (b) down-ramp rates, $R'_d(l)$.

- There is a strong correlation between the average normalised ramp rate and the duration of the ramp

- The majority of ramp durations are less than 15 hours with a median value of 8.25 hours for ramp-up events and 8.5 h for ramp-down events

- There is a significant spread in ramp rates of duration between two and 15 hours

- For ramps of duration greater than around 12 hours, ramp rates tend to decrease

- There are very few ramp events with a duration of longer than a day (24 hours)

The features described above are logical when considering the nature of the events driving wind ramps which are generally localised in nature and rarely lasting longer than a day such as the passage of a weather front, a sea breeze (Steele et al., 2015) 160 or a low-level jet (Nunalee and Basu, 2014; Kalverla et al., 2019).

## 6.2 Diurnal and seasonal dependency

We also investigate whether the ramp rates show a diurnal or seasonal dependence. To do this, we classify ramp-up and ramp-down rates based on their duration: ramps $\leq 2$ hrs are classified as short duration; ramps within the range $2 - 15$ hrs are classified as medium duration; and long duration ramps are assumed to be $\geq 15$ hrs. This classification is somewhat arbitrary, 165 but is broadly based on the discussion in Section 1. The results are shown in Figure 8. It can be seen that medium duration ramps in particular show a strong diurnal cycle with a higher frequency of ramp-up events in the afternoon and ramp-down events in the morning. This is true to a lesser extent for the long duration ramps. There is no discernible diurnal pattern in the short duration ramps. By contrast, there is no clear cycle at any scale across the different months of year. The observed

diurnal variation in medium and long duration wind power ramps is consistent with the pattern of average diurnal generation observed as seen in Figure 8(e) which is strongly influenced by mesoscale effects such as low-level jets, land-sea breezes and thermally driven entrainment from aloft due to the relatively close proximity of the Belgian offshore wind farms to the coast. Low level jets are known to be more prevalent during the evening hours at the location of some of the Belgian offshore wind farms (Kalverla et al., 2019) which may contribute to the increase in power generation observed during this period.

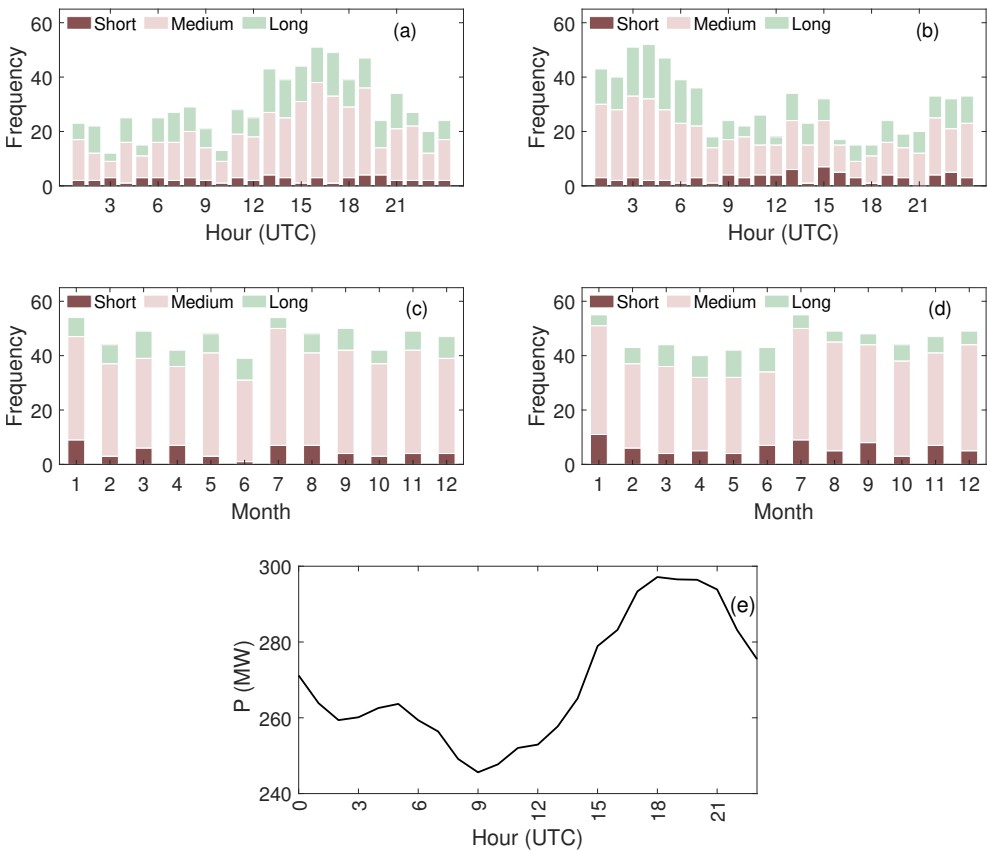

**Figure 8.** Diurnal and monthly frequency ramps events for short, medium and long duration: (a) diurnal ramp-up events; (b) diurnal ramp-down events; (c) monthly ramp-up events; (d) monthly ramp-down events; (e) diurnal variation of wind power.

These results are based on one dataset for a limited two-year period. Clearly, further work is necessary to investigate the generality of the observations above. However, this short investigation does illustrate how wavelets can be used to investigate ramps rates, their duration and prevalence.

## 7  Ramp detection during a period of change in installed power capacity

As a further illustration of the utility of the ramp detection methodology, we apply it to a different time series of wind power data where the installed capacity increases with time. Such a change can be problematic where ramps are defined using a minimum change in wind power output $\Delta P$ over a time $\Delta t$.

In this section, we use the measured hourly wind power data from the Swedish SE1 price region for the period 2000–2001. Figure 9(a) shows an increasing trend in the production of wind power which reflects an increase in installed capacity over the two-year period. This trend is also clearly observed in the continuous wavelet coefficient of the power values (P) shown in Figure 9(c). Note that in this case, we have deliberately *not* normalized the data to show how the method can be applied when the installed capacity is non-stationary. If the wind power data over the period are randomly shuffled, then clearly the trend is no longer visible in either the surrogate series seen in Figure 9(b) or its continuous wavelet transform coefficients $W^*$ as depicted in Figure 9(d).

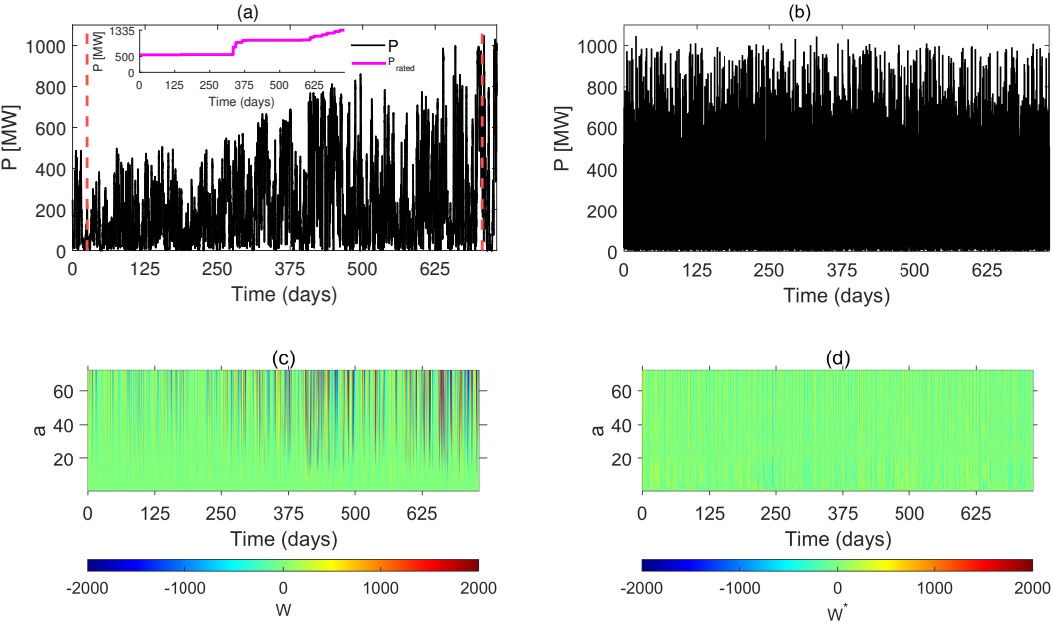

**Figure 9.** (a) hourly wind power production (P) and rated power ($P_{rated}$) for the aggregated Swedish Energy price region SE1 for 2000–2001. The dash red lines represent two periods studied in further detail (see Figure 9); (b) a randomly shuffled hourly values based on the time series in top left; (c) the continuous wavelet transform coefficient of the values in (a); and (d) the continuous wavelet transform coefficient of the randomly shuffled values in (b).

We then focus on two periods of data shown by dashed lines in Figure 9(a). In the first period, 01 Jan 2000 to 25 Jan 2000, installed capacity was around 500 MW. In the second period, 06 Dec 2001 to 31 Dec 2001, installed capacity had increased to around 1300 MW. The difference in magnitude of the $W$ values can clearly be seen in Figure 10(a) for the first

period and Figure 10(b) for the second. Then, using the entire period 2000–2001 and the method of surrogates to fix the 10% discrimination level following the same methodology as described above, we calculate the power ramps, $R(t)$ for these two periods. Although the values of $R(t)$ are of a different magnitude for the two periods, it can be seen that there is no discernible difference in the ability to detect ramps in the first period, Figure 10(c) or the second period, Figure 10(d).

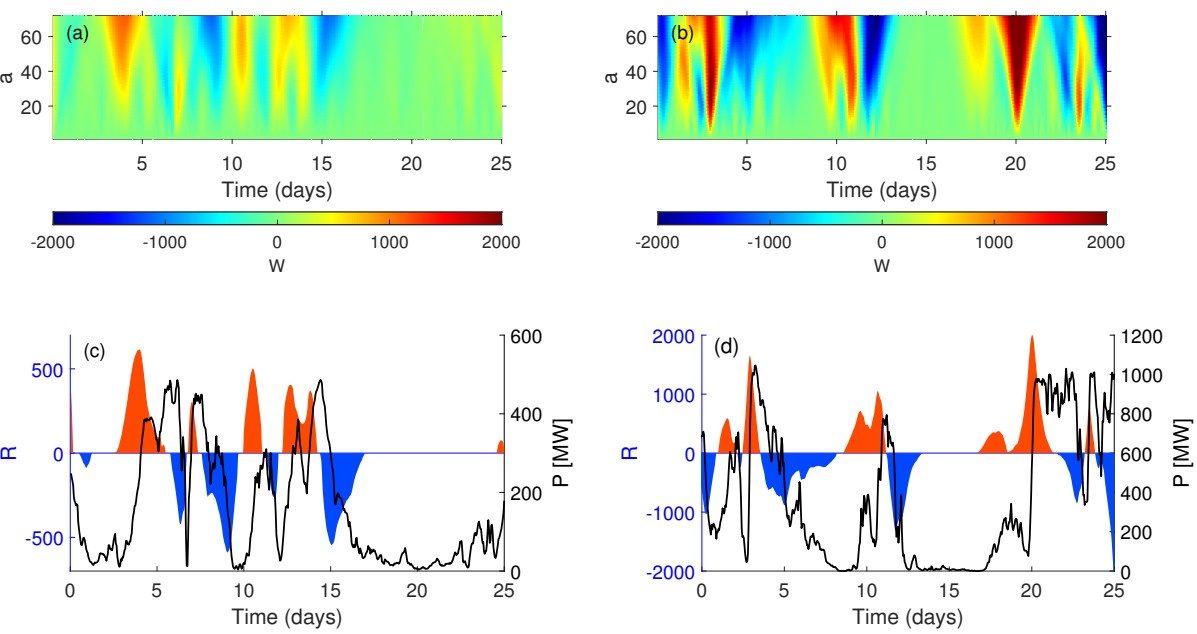

**Figure 10.** Wavelet coefficients and ramps based on wind power for the aggregated Swedish energy price region SE1 for two periods, 01 Jan 2000 to 25 Jan 2000 and 06 Dec 2001 to 31 Dec 2001; (a) continuous wavelet transform coefficient for the period 01 Jan 2000 to 25 Jan 2000; (b) continuous wavelet transform coefficient for the period 06 Dec 2001 to 31 Dec 2001; (c) power ramps for the period 01 Jan 2000 to 25 Jan 2000; and (d) power ramps for the period 06 Dec 2001 to 31 Dec 2001.

## 8    A comparison between the wavelet-surrogate and min-max ramp detection methods


Finally, we compare our wavelet-surrogate (WS) method with an existing ramp detection method known as the min-max method (Bianco et al., 2016). The min-max ramp detection method makes use of a sliding window of a given window length (WL) (in time steps) and considers the change in power during that window defined as the difference between the maximum and minimum power within the window. If this change in power is greater than a defined threshold (TH) then a ramp is deemed to have occurred. Clearly, the sensitivity of this method depends on the chosen values of two parameters, namely WL and TH in contrast to our WS methodology which depends on a discrimination level. In this study, we compare the two methods in detecting the number of up and down ramps events. This comparison is made for both the Belgian and Swedish wind power data-sets. Table 2 shows the number of ramp-up and ramp-down events that are detected using the WS and min-max methods


for different time periods of the two wind power data-sets as a function of different parameter values used by each method.

For the WS method, we quantify the sensitivity of ramp detection using the 10% (WS10), 5% (WS5), 2% (WS2) and 1% (WS1) discrimination levels. In the case of the min-max method, we use a combination of window lengths (WL1 = 8 and WL2 = 12) and threshold levels (TH1 = 0.3 and TH2 = 0.4). For the 15-minute Belgian offshore wind power, these WL values correspond to WL1 = 2 h and WL2 = 3 h, whereas for the hourly Swedish onshore wind power data, they correspond to WL1 = 8 h and WL2 = 12 h. Note that the actual Swedish wind power data values are used for the WS algorithm while the data are

normalized by time-varying installed power capacity for the min-max method (see Figure 9(a)). In general, the number of ramp events detected by the WS method is greater than for the min-max method. The number of events detected by the WS method generally reduces as the discrimination percentage value gets smaller. The exception to this is for the T1 and T2 periods where a slight increase is seen at the 1% discrimination level when the overall event count is low. This is due to the splitting of single ramp events into two events, an example of which can be seen around day four in Figure 5(d). Similarly, the min-max method

detects fewer ramp events as TH is increased. By contrast, increasing the WL value does not show such a clear trend in the number of ramp events detected. In addition, the min-max method seems more sensitive to the range of WL and TH values used than for the range of discrimination values used for the WS method.

**Table 2.** Comparison of the wavelet-surrogate (WS) and min-max ramp detection methods. The number of ramp-up and ramp-down events are shown in bold and in parenthesis, respectively. Periods T1–T4 are given in Table 1, T5 is the Belgian offshore wind power data for the entire period 2015–2016 and SE1 is the hourly Swedish wind power data for the period 2000–2001.

| Times | WS | | | | min-max method | | | |
|---|---|---|---|---|---|---|---|---|
| | WS10 | WS5 | WS2 | WS1 | WL1, TH1 | WL1, TH2 | WL2, TH1 | WL2, TH2 |
| T1 | **6**(6) | **6**(5) | **4**(4) | **6**(4) | **4**(2) | **1**(2) | **4**(3) | **4**(3) |
| T2 | **11**(9) | **8**(8) | **8**(7) | **7**(8) | **6**(4) | **4**(2) | **7**(5) | **5**(3) |
| T3 | **16**(15) | **14**(13) | **13**(12) | **12**(4) | **15**(13) | **10**(6) | **14**(18) | **11**(10) |
| T4 | **10**(12) | **10**(10) | **6**(8) | **5**(6) | **2**(5) | **1**(2) | **3**(6) | **1**(4) |
| T5 | **567**(564) | **498**(487) | **434**(430) | **397**(390) | **426**(386) | **214**(181) | **527**(467) | **294**(260) |
| SE1 | **237**(244) | **209**(220) | **193**(194) | **179**(182) | **185**(179) | **89**(85) | **224**(224) | **132**(139) |

## 9   Conclusions

The detection of wind power ramps is a challenge in terms of how to characterise their magnitude and duration and how to

discriminate a ramp from incoherent stochastic fluctuations in wind power. In this paper, we have presented a relatively simple methodology based on a wavelet transform and the use of surrogates to discriminate and extract ramp events. Using wind power data from the Belgian offshore wind farm cluster, we have illustrated the application of the methodology and have shown that a ten-day period is sufficient to discriminate coherent ramp events from incoherent fluctuations. We show the utility of the technique in characterising the distribution of ramp rates and their duration, seasonal-diurnal variation for the Belgian offshore

cluster. In addition, we have shown how the methodology can be used to detect wind power ramps when installed capacity increases with time using Swedish onshore wind power data as an example. Lastly, we compare our ramp detection algorithm with the min-max method contrasting their sensitivity to parameter choice. Further work is required to apply the methodology to a broader range of sites and for longer periods to investigate the prevalence of different ramp rates and their duration. It might be expected that depending on the climatology of the site that this could differ; on the other hand, consistent trends may

be apparent which could help operators in accommodating fluctuations within an integrated power system.

*Data availability.* The data-sets are freely available at (Elia, 2020) and (EEM20, 2020)

*Author contributions.* This research was carried out by Bedassa R. Cheneka under the supervision of Simon J. Watson and Sukanta Basu.

*Competing interests.* The authors declare that they have no conflict of interest.

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
