# Peer review of "A simple methodology to detect and quantify wind power ramps"

_Wind Energy Science, 2020_

## Referee Comment (RC1) · Anonymous Referee #1 · 28 May 2020

Review of: wes-2020-64

Title: "A simple methodology to detect and quantify wind power ramps" Author(s): Bedassa R. Cheneka, Simon J. Watson, and Sukanta Basu

General Comments:

Ramp events are rapid changes of wind power production over short period of times and grid operators have to be prepared to switch between renewable and other forms of energy during these rapid changes in wind power availability. The definition of ramp events is not unique and this manuscript describes a methodology that uses a wavelet transform to discriminate ramp events above stochastic wind variations. The subject of the study is suitable to Wind Energy Science. Overall the manuscript reads well and

the methodology is adequately described.

Figures are OK, although some improvements have to be performed, as suggested in the specific comments below.

This Referee thinks that since other procedures, also referenced in the manuscript, have been introduced in other studies the authors could explain better what are the benefits of this methodology versus the others and maybe even compare the results of ramp identification when using other methodologies against this one. For instance, the Min-Max method introduced in Bianco et al. 2016 is easily reproducible and it could be employed with many different ramp definitions on the 10 days data-set used in this manuscript, marking positively all the times when any ramp is identified, and comparing with the output of the wavelet transform method introduced here.

Also, the authors claim that the disadvantages of other methods are that they rely on specific thresholds, but it seems that this procedure also rely on some decision on thresholds (i.e. the authors decide to go with a 10% rejection level, instead of a 5%, 2% or 1%). This threshold could be dependent on the data-set, as I imagine that aggregating over more turbines would smooth out the wind power time series.

Finally, since the data-set used in Section 6 is longer (2 years) that the one used in the previous sections to introduce the methodology (10 days), this section could be expanded to include more quantitative statistical results, maybe also looking at daily, or seasonal distribution of ramp events, for instance.

Specific Comments:

Fig. 1: You need to stretch the x-axis of the bottom panel to match the length of the upper panel's one, as it is difficult now to see how the large values of W in the lower panel correspond to the up- or down-ramps visible in the upper panel.

Page 3, lines 83-84 and page 4, lines 85-86: Referring to Fig. 2 you mention "top left plot" and "bottom right plot", which makes me thing there are 4 panels in your figure,

but I can only see a left and right panel.

Fig. 2: Please introduce what P and P' are in the figure caption.

Fig. 3: Change label to explain that left panels are for 10% and 5% rejection levels, and right panels for 2% and 1%. Also, introduce what is R on the left-y-axis.

Fig. 4: "the colour scale is blue (ramp-up events), red (ramp-down events)". Isn't it the opposite of what you are saying?

Fig. 5: Could you try to keep consistency with previous figures in colors identifying up- and down-ramp events?

Page 9, Lines 146-147: "though the correlation is stronger for the ramp-up events than the ramp-down events". Can you provide statistical values for this statement?

Section 6: I think in general this section could be expanded to include more quantitative statistical results as well as analysis of daily and seasonal distribution of up- and down-ramp events, for instance.

Fig. 7: How are the durations binned?

Technical Corrections:

Page 2, line 40: Replace "was used detect ramps" with "was used to detect ramps"

---

## Referee Comment (RC2) · Anonymous Referee #2 · 18 Jun 2020

This manuscript presents a simple methodology for detecting wind power ramps using a wavelet transform. The narrative is written in a concise manner, with a proper literature review and a fair discussion of other ramp detection methods. There are, however, issues regarding how ramps are defined and the whether the training data is sufficient to justify the conclusions reached. The manuscript suffers from a lack of further (or deeper) analysis into ramps of more interest–those of shorter duration.

More specifically, the following issues should be addressed prior to acceptance for publication:

1. The authors use terms such as "rapid", "significant", "relatively short time", "short-duration", and "longer duration" without quantifying what these terms mean in providing context for their ambiguous use of the term "ramp."

2. It is unclear what definition, if any, the Belgian transmission system operator uses to identify a ramp event.

3. Are the 10-day training periods sufficient? Are these periods representative of ramp event distributions and the weather responsible for ramps in this part of the world? It is somewhat unclear what is meant by "one calendar year of values"–I assume these were the data used to generate the distributions in Figure 7.

4. I am not sure what the value of the longer-duration ramp events (median value 8.5 hours) would be for a generation facility of this size. Of greater interest to most utilities are ramp events that occur on truly short time sales, such as those analyzed in the cited papers (e.g. a 20% or more change in output over a period of an hour). There certainly appears to be a high frequency of these events as illustrated in Figure 7, but little discussion. How predictable would these be with this method? Would there be false alarms?

---

## Author Comment (AC1) · 18 Aug 2020

[journal abbreviation, manuscript]copernicus

Thank you for your detailed comments. Following your comments and feedback we have added some additional results and explanation to improve the quality of the paper. The most important changes to the paper include:

- We have made an analysis of the seasonal and diurnal variation of the ramps for the offshore data.

- We have analysed an additional data-set (Swedish onshore wind farm data)

[Figure]

which allows us to show how it can be used when the installed capacity of the fleet changes during the time series being analysed.

- We have compared our method with with the min-max ramp identification method of Bianco et al. 2016.

We have responded to the referee comments by including our answers below the original comments. Our answers are preceded with the following labels:

RC = referee's comments
AC = authors' comments

**General comments**

RC: Ramp events are rapid changes of wind power production over short period of times and grid operators have to be prepared to switch between renewable and other forms of energy during these rapid changes in wind power availability. The definition of ramp events is not unique and this manuscript describes a methodology that uses a wavelet transform to discriminate ramp events above stochastic wind variations. The subject of the study is suitable to Wind Energy Science. Overall the manuscript reads well and the methodology is adequately described. Figures are OK, although some improvements have to be performed, as suggested in the specific comments below.

This Referee thinks that since other procedures, also referenced in the manuscript, have been introduced in other studies the authors could explain better what are the benefits of this methodology versus the others and maybe even compare the results of ramp identification when using other methodologies against this one. For instance, the Min-Max method introduced in Bianco et al. 2016 is easily reproducible and it could be employed with many different ramp definitions on the 10 days data-set used

in this manuscript, marking positively all the times when any ramp is identified, and comparing with the output of the wavelet transform method introduced here.

AC : Thank you for this feedback. We have expanded the scope of the paper to show the advantages of the method (including the analysis of a data-set where installed capacity changes). We have compared the wavelet transform method with the min-max method and contrasted sensitivity to parameter choices.

RC: Also, the authors claim that the disadvantages of other methods are that they rely on specific thresholds, but it seems that this procedure also rely on some decision on thresholds (i.e. the authors decide to go with a 10% rejection level, instead of a 5%, 2% or 1%). This threshold could be dependent on the data-set, as I imagine that aggregating over more turbines would smooth out the wind power time series.

AC: We agree that the statistical discrimination level is a matter of choice, but we believe that this approach is more robust and less sensitive to approaches which rely on specific power level or time-scale changes, particularly where a data-set is non-stationary, e.g. where installed capacity increases. We have added some additional explanation and data analysis (the Swedish data and the comparison with the min-max method) to try and better explain the advantages of the method.

RC: Finally, since the data-set used in Section 6 is longer (2 years) that the one used in the previous sections to introduce the methodology (10 days), this section could be expanded to include more quantitative statistical results, maybe also looking at daily, or seasonal distribution of ramp events, for instance.

AC: We have expanded the analysis to look at diurnal and seasonal variation for the Belgian offshore wind farm cluster.

**Specific comments**

RC: Fig. 1: You need to stretch the x-axis of the bottom panel to match the length of the upper panel's one, as it is difficult now to see how the large values of W in the lower panel correspond to the up- or down-ramps visible in the upper panel.

AC: This figure has been changed.

RC: Page 3, lines 83-84 and page 4, lines 85-86: Referring to Fig. 2 you mention "top left plot" and "bottom right plot", which makes me thing there are 4 panels in your figure, but I can only see a left and right panel.

AC: Figures have been revised and each sub-figure is now referred to as (a), (b), (c), etc.

RC: Fig. 2: Please introduce what P and P' are in the figure caption.

AC: We have made this explicit in the caption.

Rc: Fig. 3: Change label to explain that left panels are for 10% and 5% rejection levels, and right panels for 2% and 1%. Also, introduce what is R on the left-y-axis.

AC: We have changed the plots in this figure and added text in the paper to make things clearer.

RC: Fig. 4: "the colour scale is blue (ramp-up events), red (ramp-down events)". Isn't it the opposite of what you are saying?

AC: This has been corrected.

RC: Fig. 5: Could you try to keep consistency with previous figures in colors identifying up and down-ramp events?

AC: We have changed this to make things more consistent.

RC: Page 9, Lines 146-147: "though the correlation is stronger for the ramp-up events than the ramp-down events". Can you provide statistical values for this statement?

AC: As we have expanded the resolution of this figure, this is no longer as obvious so we have removed the comment about the correlation being stronger for the ramp-up events.

RC: Section 6: I think in general this section could be expanded to include more quantitative statistical results as well as analysis of daily and seasonal distribution of up- and downramp events, for instance.

AC: We have expanded the analysis to look at seasonal and diurnal trends.

RC: Fig. 7: How are the durations binned?

AC: This is hourly. We have added this in the text.

**Technical corrections**

RC: Page 2, line 40: Replace "was used detect ramps" with "was used to detect ramps"

[Figure]

AC: We have amended this sentence.

---

## Author Comment (AC2) · 18 Aug 2020

[journal abbreviation, manuscript]copernicus

Thank you for your detailed comments. Following your comments and feedback we have added some additional results and explanation to improve the quality of the paper. The most important changes to the paper include:

- We have made an analysis of the seasonal and diurnal variation of the ramps for the offshore data.

- We have analysed an additional data-set (Swedish onshore wind farm data)

[Figure]

which allows us to show how it can be used when the installed capacity of the fleet changes during the time series being analysed.

- We have compared our method with with the min-max ramp identification method of Bianco et al. 2016.

We have responded to the referee comments by including our answers below the original comments. Our answers are preceded with the following labels:

RC = referee's comments
AC = authors' comments

**General comments**

RC: This manuscript presents a simple methodology for detecting wind power ramps using a wavelet transform. The narrative is written in a concise manner, with a proper literature review and a fair discussion of other ramp detection methods. There are, however,issues regarding how ramps are defined and the whether the training data is sufficient to justify the conclusions reached. The manuscript suffers from a lack of further (or deeper) analysis into ramps of more interest–those of shorter duration.

AC: We have significantly increased the depth of the analysis to consider diurnal and seasonal variation at the offshore site, the analysis of additional data from an onshore site and a comparison with another ramp detection method.

RC: More specifically, the following issues should be addressed prior to acceptance for publication:

RC: 1. The authors use terms such as "rapid", "significant", "relatively short time", "shortduration", and "longer duration" without quantifying what these terms mean in providing context for their ambiguous use of the term "ramp."

AC: Thank you for your feedback. We agree that these terms are subjective. We have now classified short, medium and long duration ramps with specific timescales and based this on published work elsewhere. The classification is described in the first section and used in the extended analysis later in the paper.

RC: 2. It is unclear what definition, if any, the Belgian transmission system operator uses to identify a ramp event.

AC: We are not aware that the Belgian TSO uses a specific classification. In this work, the intention was merely to use the Belgian data as one example, but that this could be applied to other data-sets. As mentioned in the reply above, we have now attempted to classify ramps in terms of time periods of relevance to system operators in general based on published work elsewhere.

RC: 3. Are the 10-day training periods sufficient? Are these periods representative of ramp event distributions and the weather responsible for ramps in this part of the world? It is somewhat unclear what is meant by "one calendar year of values"–I assume these were the data used to generate the distributions in Figure 7.

AC: We have now analysed sensitivity of the method to the length of the training period.

RC: 4. I am not sure what the value of the longer-duration ramp events (median value 8.5 hours) would be for a generation facility of this size. Of greater interest to most utilities are ramp events that occur on truly short time sales, such as those analyzed in the cited papers (e.g. a 20% or more change in output over a period of an hour). There certainly appears to be a high frequency of these events as illustrated

in Figure 7, but little discussion. How predictable would these be with this method? Would there be false alarms?

AC: As mentioned above, we have tried to provide some context in terms of the relevance of different ramp durations based on other published work. We think there may also be some confusion by the reviewer as nowhere do we suggest that we are predicting ramps. The paper is only concerned with identifying ramps from observed data.